# The Use of Collections of Artificial Neural Networks to Improve the Control Quality of the Induction Soldering Process

**DOI:** 10.3390/s21124199

**Published:** 2021-06-18

**Authors:** Anton Vladimirovich Milov, Vadim Sergeevich Tynchenko, Sergei Olegovich Kurashkin, Valeriya Valerievna Tynchenko, Vladislav Viktorovich Kukartsev, Vladimir Viktorovich Bukhtoyarov, Roman Sergienko, Viktor Alekseevich Kukartsev, Kirill Aleksandrovich Bashmur

**Affiliations:** 1Information-Control Systems Department, Institute of Computer Science and Telecommunications, Reshetnev Siberian State University of Science and Technology, 660037 Krasnoyarsk, Russia; antnraven@ieee.org (A.V.M.); scorpion_ser@mail.ru (S.O.K.); 051301@mail.ru (V.V.T.); vlad_saa_2000@mail.ru (V.V.K.); vladber@list.ru (V.V.B.); 2Department of Technological Machines and Equipment of Oil and Gas Complex, School of Petroleum and Natural Gas Engineering, Siberian Federal University, 660041 Krasnoyarsk, Russia; vakukartsev@sfu-kras.ru (V.A.K.); bashmur@bk.ru (K.A.B.); 3Machine Learning Department, Gini Gmbh, 80339 Munich, Germany; roman@gini.net

**Keywords:** industrial sensors, pyrometers, non-contact temperature measurement, measurements reliability, induction brazing, intelligent technologies, artificial neural networks, waveguide paths

## Abstract

In industries that implement the technology of induction soldering, various sensors, including non-contact pyrometric ones, are widely used to control the technological process. The use of this type of sensor implies the need to choose a solution that is effective in different operating conditions in terms of the accuracy of the data obtained and the reliability of the measurement equipment and duplication in case of a failure. The present article discusses the development of intelligent technology based on a collection of artificial neural networks, which allows a number of problems associated with technological process control when using pyrometric sensors to be solved: assessing the quality of measurements, correcting measurements when non-standard errors are detected, and controlling the process of induction heating in the absence of reliable readings of the measurement instruments. The collection of artificial neural networks is self-configuring with the use of multicriterion genetic algorithms. The use of the proposed intelligent technology made it possible to improve the control quality of the technological process of the induction brazing of waveguide paths of spacecraft: the overregulation was decreased from 0–20 to 0, and the difference in the heating temperatures of the elements of the brazed waveguide assembly was decreased from 20–100 to 0–10. In addition, the overall process duration decreased and became more stable. When using the classical control technology, the time varied in the range of 20–60 s; when using the proposed technology, it stabilized in the range of 30–35 s.

## 1. Introduction

In the production of waveguide paths for spacecraft (SCWP), induction brazing (IB) technology is most often used by the enterprises of the aerospace industry to create high-quality one-piece connections [1,2]. The use of such a high-tech method is due to the high requirements in terms of the quality of the products [3]. The use of IB when connecting the assembly elements of the waveguide path allows for the finished product to comply with the requirements relating to radio technical characteristics and it reduces the mass-dimensional indicators, which, in turn, reduces production costs.

The control of the induction brazing technological process (TP) occurs under conditions of uncertainty, which are caused by the presence of many negative factors associated mainly with the use of non-contact temperature measurement sensors [4].

One of the main modern directions in the automation of industrial production is the implementation of automated systems within the framework of the Industry 4.0 concept. This concept presupposes the creation and implementation of new cyber-physical systems in various industries, which, using intelligent information processing methods, allows the quality of technological process control to be improved. This, in turn, improves the quality of products [5].

The use of intelligent methods makes it possible to raise the reliability of information [6,7] obtained from the heating zone in order to estimate the errors of measurement instruments and form an adequate control of the TP, which allows its accuracy and repeatability to be improved [8].

The purpose of this work is to present an intelligent technology for controlling the IB process of waveguide paths of spacecraft (SCWP). The considered intelligent technology is designed to ensure all stages of the TP IB, from the initial setting of the technological parameters to the dynamic switching of the operating modes of a specific system, are implemented on the basis of the proposed technology, which controls the current state of the TP. The essence of the proposed technology lies in the functioning throughout the entire technological process of a trained artificial neural network that analyzes information on the process temperature using temperature measurement instruments (non-contact pyrometric sensors), the result of which is the determination of the presence of incorrect information as well as the possibility of correcting this information using another pre-trained artificial neural network.

The use of the intellectual technology presented in this study will significantly improve the control quality of TP IB of SCWP by reducing costs associated with the features of TP and contactless temperature measurement sensors.

## 2. Materials and Methods

The main principle of Industry 4.0 is the massive introduction of information technologies in the industry, the large-scale automation of business processes, and the spread of artificial intelligence [9]. To date, various sensors have been used by enterprises in the production process; for example, contactless pyrometric sensors are used in the manufacturing of SCWP in the brazing process [10]. Within the framework of the “reliability” theory, appropriate sensors are selected that meet the specified requirements (MTBF, noise immunity, etc.) and the requirements of the technological process (sensor type, temperature measurement range, connection port type, etc.). However, in order to prevent failures in the operation of the system as a whole and the sensors used in particular, in Russia, the same theory of “reliability” is used [11], which has been shown to be effective but is outdated and has shown complete inconsistency in some cases (for example, the absence of the possibility of eliminating errors when using trimming resistors to control the electron beam current during welding or, for example, an uncontrollably variable polling rate from pyrometric sensors when using a COM port connection connected directly with the features of the COM port and the board on which it is located) [12,13]. To prevent such problems, in this work, the development and use of an intelligent system are proposed, using the example of a specific production, which allows for the collection and storage of information received from sensors on a separate server in real time and also the monitoring of the technology, and in the event of a partial or complete failure of all of the sensors, the control is switched to automatic mode, and the technological process is completed after the finished product is received. The technology developed by the authors in this study is intended to improve the control quality of the process of the induction brazing of waveguide paths of spacecraft based on the use of intelligent methods.

The process of the induction brazing of waveguide paths is fast flowing [14], and the average total duration is within 20–60 s. In addition, the technology of induction packs in the enterprises of the rocket and space industry has been used for a long time, which by now has made it possible to accumulate a large amount of experimental data. Due to the abovementioned two features of the technological process under consideration, this work uses teaching “with a teacher”.

In the previous works [15,16], the authors investigated the application of the following data analysis methods to solve the problem of controlling the technological process of induction soldering: decision trees [16], artificial neural networks [15,16], fuzzy logic [15,16], and neuro-fuzzy controller [15]. According to the results of the experiments, artificial neural networks (ANN) showed the best accuracy. This led to the choice of such a technology for solving the problems posed in this study, the datasets in which are similar to the data in works [15,16].

The supervised learning of an artificial neural network is a multidimensional optimization problem [17]. To solve this problem, it is necessary to find a set of weighting coefficients that minimize the error function. To solve this problem, gradient and stochastic optimization methods are suitable.

Mathematical models based on artificial neural networks have been successfully used to solve various problems. Examples of successful use of artificial neural networks are:the development of a model for predicting the reliability of complex software systems [18];the use of ANN to solve environmental problems [19];the recognition of objects by the structure of a material [20];the identification and diagnostics of technical objects [21];the control of the technological process of thermochemical dehydration [22];the management of an energy converter [23];the control of the parameters of technological processes at thermal power plants [24];the use of ANN to solve energy problems [25,26].

### 2.1. Intelligent Technology

The developed technology is used to control the process of the induction brazing of waveguide paths. The functional diagram of the installation used in the considered technological process is shown in Figure 1 and includes the following components:a generator of induction heating;a matching device;a set of inductors;pyrometric temperature sensors;an electromechanical drive.

The application of the developed approach was carried out for the process of the induction brazing of a pipe-flange waveguide assembly. The experiments were carried out using the equipment shown in Figure 2, which have the following characteristics:a remote control;a voltage source up to 10 V, output current up to 150 A, and power up to 15 kW;an autonomous cooling system;a pyrometric temperature stabilization circuit; a manipulator.

The installation consists of the following elements: a control unit; an IPPC-9171G-07BTO industrial computer; a PCI-1710 interface board; two pyrometric temperature sensors, which are widely used for non-contact temperature measurements [27,28]; an induction heating generator; a matching device; an electromechanical drive; and an inductor. The temperature reading is carried out using various elements of the SCWP assembly: one pyrometer (“Pyrometer 1”) measures the temperature of the SCWP pipe, and the other (“Pyrometer 2”) measures the temperature of the SCWP flange or coupling.

To measure the heating temperature of the assembly elements of the waveguide path, the AST 250 pyrometers shown in Figure 3 are used.

These pyrometers are designed for the high-quality, non-contact temperature measurement of parts heated in harsh industrial environments. The pyrometers operate at ambient temperatures up to 250 °C without additional cooling. Since the fiber optic cable and optical head do not contain electronic components, the AST 250 pyrometers can be used in the presence of electromagnetic interference in the working area. The temperature measurement range of the pyrometers is in the range of 300 to 2500 °C. The minimum reaction time of the pyrometers is 2 milliseconds. The pyrometers can be connected via RS-232 or RS-485 interfaces. The pyrometer is aimed at the temperature measurement zone using a laser indicator.

The induction heating of the waveguide is controlled via two channels: the channel for controlling the power setting of the induction heating generator and the channel for controlling the position of the SCWP assembly relative to the inductor window.

The channel for controlling the position of the SCWP assembly consists of an electromechanical drive connected to the SCWP assembly. According to the state of the position of the SCWP assembly in the inductor window, a control action is formed using an industrial computer. The control action is transmitted via the interface board to the electromechanical drive. The power setting control channel consists of an induction heating generator and a matching device. The signal controlling the power of the induction heating generator is fed from the industrial computer through the interface board to the generator. The matching device provides the transfer of energy from the generator to the inductor, carrying out the induction heating of the SCWP assembly.

Through our experimental studies of the technological process of the induction brazing of SCWP, it was found that the programmed control of the heating power allows one to control only one parameter—the temperature of the waveguide pipe. The heating temperatures of the elements of the SCWP assembly can vary significantly. The difference can be up to 30–70 °C.

The selection of the correct position of the SCWP assembly relative to the inductor window makes it possible to reduce the difference in heating temperatures when the solder melting temperature is reached. However, given that pipes with a difference in thickness of up to 20% can be used in SCWP assemblies, it is not possible to completely eliminate the discrepancy between the heating temperatures of the SCWP assembly elements.

It is possible to improve the control quality of the SCWP induction brazing process by introducing a second control loop. The first circuit is required for controlling the power setpoint. The second loop is necessary for controlling such a parameter as the position of the SCWP assembly relative to the inductor window [29].

The principle of the developed intelligent technology lies in the functioning throughout the entire brazing process of a pretrained artificial neural network [30,31,32,33] (ANN_ident_), which analyzes information about the temperature of the SCWP assembly elements from the measurement instruments. The technology makes it possible to determine the presence of incorrect information about TP, as well as the possibility to correct it using another pre-trained artificial neural network (ANN). A typical scenario of the proposed technology is shown in Figure 4.

At the moments of time T = (t_1_, ..., t_n_), one of the possible events occurs in the TP of the IB of SCWP, leading to errors in the operation of the control system based on classical control algorithms, particularly a PID controller. An example of the process is shown in Figure 4.

The list of possible events can be reduced to the following types:the emergence of a standard error, the correction of which is not necessary;the occurrence of an abnormal error that can be corrected;a lack of readings from one of the pyrometric sensors;a lack of readings from all pyrometric sensors.

In the section up to t_0_ (the heated product reaches the temperature corresponding to the lower limit of the pyrometer measurements), temperature control is not performed; therefore, for control, the use of an intelligent algorithm—a pre-trained ANN for control (ANN_control_)—is proposed.

In this case, there is a division into low-temperature and high-temperature pyrometers. The use of pyrometric sensors with a wide range of measured temperatures is technically ineffective due to the high requirements concerning the accuracy of temperature measurements of the process of the induction brazing of waveguide paths of spacecraft and the field of information retrieval. It is also economically impractical due to the high cost.

Due to this feature of the measurement instruments, high-temperature pyrometers are used in the automation of the induction brazing process, which measure temperatures in the range of 300 to 1800 °C. Accordingly, there is a section in the technological process of heating the product (up to 300 °C) when the temperature is not controlled; therefore, in this section until t_0_ (the heated product reaches the temperature corresponding to the lower limit of the pyrometer measurement), the use of an intelligent algorithm for control—specifically, a pre-trained artificial neural network—is proposed.

In the section from t_0_ to t_1_, normal temperature measurements of the technological process take place, and classical algorithms are used for control, particularly a PID controller.

At time t_1_, due to the evaporation of the flux, incorrect information begins to flow from one of the pyrometers due to a change in the emissivity of the material at the measurement point. In this situation, the intelligent system corrects the readings from the pyrometers using another pre-trained ANN for correction (ANN_corr_), and classical algorithms are used for control in the section from t_1_ to t_2_ (the time when the measurements are restored).

At time t_3_, one of the pyrometers fails. Since there are no readings from one of the pyrometric sensors, the intelligent system starts predicting measurements using another pre-trained ANN (ANN_predict_). At time t_4_, two pyrometers fail. In this situation, the system goes into an intelligent control mode, which is similar to the mode of operation on the site until time t_0_.

### 2.2. Algorithm of Induction Brazing Intelligent Control

The block diagram of the control algorithm of the developed intelligent technology is shown in Figure 5. At the initial stage of the intelligent control of the IB of SCWP process, data are obtained from pyrometric temperature sensors. If both sensors provide data on the process temperature, then errors are identified. In the absence of errors or in the presence of standard errors, the data are transmitted further for TP control based on classical control algorithms. In the presence of non-standard errors, they are corrected using further data transmission for control based on classical algorithms. In the absence of readings from one of the pyrometers, the data from the second sensor are determined based on the forecast obtained using the ANN_predict_, while the control is also based on classical control algorithms. If there are no data from both pyrometric sensors, then control is carried out based on ANN_control_. A sign of stopping the control process is the expiration of the stabilization time of the temperature of the product, which is produced at the final stage of the technological process of the induction brazing of the SCWP.

### 2.3. Artificial Neural Networks for Induction Brazing Intelligent Control

Having considered the problem of building a neural network in general in Section 2.2, we can move on to a specific case, namely, the task of developing an intelligent system for the induction brazing process. We formulate the problem of identifying non-standard errors as follows.

A leaky linear rectifier (Leaky ReLU) was chosen as the activation function for each neural network. A batch normalization layer is also used for the input data. Batch normalization applies a transformation that maintains the mean output close to 0 and the output standard deviation close to 1 [34]. The dropout layer randomly sets input units to 0 with a frequency of rate at each step during training time, which helps to prevent overfitting. Inputs not set to 0 are scaled up by 1/(1 − rate) such that the sum over all inputs is unchanged. We assume that we have the following input and output data:T_1_, …, T_n_—input data, which are time-series data from the technological process of brazing, i.e., the temperature of one of the elements to be brazed;Pr—output class denoting an error;Corr—output class denoting the normality of the error.

There is an unknown target mapping dependency: y*: T_1_, …, T_n_ → Pr, Corr, the value of which is known only in the training sample. It is necessary to develop a mapping algorithm capable of classifying an arbitrary object from sets T_1_, …, T_n_. The general structure of an artificial neural network for identifying errors in measurement instruments (ANN_ident_) is shown in Figure 6.

We formulate the problem of correcting non-standard errors as follows. We assume that we have the following input and output data:T_1_, …, T_m_—input data, which are time-series data from the technological process of brazing, i.e., the temperature of one of the elements to be brazed;T_m_^corr^—output value representing the corrected measurement.

There is an unknown target mapping dependency: y*: T_1_, …, T_m_ → T_m_^corr^, the value of which is known only in the training set. It is necessary to develop a an algorithm capable of classifying an arbitrary object from sets T_1_, …, T_m_. The structure of an artificial neural network for correcting errors in measurement instruments (ANN_corr_) is shown in Figure 7.

We formulate the problem of predicting measurements in the process of induction soldering as follows. We assume that we have the following input and output data:T_1_^ctrl^, …, T_m_^ctrl^—input data representing time-series data from a functioning pyrometer;T_1_^corr^, …, T_m_^corr^—input data representing time-series data from a failed pyrometer;T_m+1_^predict^—output value representing the predicted measurement.

There is an unknown target mapping dependency: y*: T_1_^ctrl^, …, T_m_^ctrl^, T_1_^corr^, …, T_m_^corr^ → T_m+1_^predict^, the value of which is known only in the training set. It is necessary to develop a mapping algorithm capable of classifying an arbitrary object from sets T_1_^ctrl^, …, T_m_^ctrl^, T_1_^corr^, …, and T_m_^corr^. The structure of an artificial neural network for predicting (ANN_predict_) values from a failed pyrometer is shown in Figure 8.

The problem of the intelligent control of the induction brazing of waveguide paths of spacecraft can be stated as follows. The following inputs and outputs are available:T_1_^p1^, …, T_n_^p1^—input data representing time-series measurements of the first pyrometer;T_1_^p2^, …, T_m_^p2^—input data representing time-series measurements of the second pyrometer;h_1_, …, h_k_—input data representing a time series of distance values from the inductor to the waveguide assembly;W_1_, …, W_k_—input data representing a time series of power setpoint values;h_k+1_—ANN output, which is the calculated value of the distance from the inductor to the workpiece;W_k+1_—ANN output representing the calculated value of the inductor power setting.

There is an unknown target mapping dependency: y*: T_1_^p2^, …, T_m_^p2^, T_1_^p2^, …, T_m_^p2^, h_1_, …, h_k_, W_1_, …, W_k_ → h_k+1_, W_k+1_, the value of which is known only in the training set. It is necessary to develop a mapping algorithm capable of classifying an arbitrary object from sets T_1_^p1^, …, T_n_^p1^, T_1_^p2^, …, T_m_^p2^, h_1_, …, h_k_, W_1_, …, W_k_. The structure of an artificial neural network for intelligent control (ANN_control_) is shown in Figure 9.

At the same time, in the initial stage of intelligent control, real temperature measurements are used, but over time, increasing incoming data represent the prediction using ANN_predict_. Likewise, the actual values of the power settings and the distance from the inductor to the workpiece are replaced with the predicted values.

The depths of windows n, m, and k represent the optimal number of measurements required to solve the assigned tasks. These values are determined empirically.

The joint use of the described pre-trained ANNs makes it possible to control the technological process of the induction brazing of waveguide paths, considering the correction of the revealed non-standard errors, and to bring the technological process to the end in the event of a failure of both contactless temperature measurement sensors.

### 2.4. Determination of the Optimal Structures of Artificial Neural Networks Used to Solve the Assigned Tasks

As a method for determining the optimal structure of an artificial neural network for solving the problems posed in this study, genetic algorithms are used. A genetic algorithm is an optimization method based on the ideas of natural evolution [35].

The way in which the problem is presented in a form suitable for applying the genetic algorithm depends on the conditions of the problem. For the purposes of this study, the parameters will be encoded with real numbers. The length of the chromosome depends on the number of optimized parameters.

#### 2.4.1. Statement of the Problem of Choosing the Best Structure of an Artificial Neural Network Using a Genetic Algorithm

The genetic algorithm for determining the best structure of an artificial neural network can be represented in the form of a block diagram (Figure 10).

The details of the execution of each stage (subprocess) can be described as follows:The creation of an initial population of chromosomes. The genotype of each of these chromosomes is the encoded parameters of an artificial neural network. This step involves the creation of a population of individuals in whose chromosomes the settings of artificial neural networks are encoded in terms of structure. Within the framework of the tasks in the chromosome, it makes sense to encode the number of hidden layers and the number of neurons in each hidden layer. For the input layers involved in the tasks, there is less variability. The number of input neurons directly depends on the depth of immersion into the sliding window. In this task, the meaningful values are the number of neurons, which is between 5 and 25. This is explained by the peculiarities of the technological process. The temperature is measured 5 times per second. The process is fast. Considering this information, as well as the average heating rate of 20–25 °C/sec, significant temperature changes requiring attention or correction, on average, occur over a period of time, which is between 2 and 5 s. Accordingly, experiments are conducted to evaluate the structures of artificial neural networks, with the number of layers corresponding to the immersion in lagged space, which is between 10 and 25 dimensions.The creation of many artificial neural networks based on the parameters encoded in the chromosomes of a population. This step involves the creation of an artificial neural network based on the phenotype data of each chromosome.Training each artificial neural network based on the available training data and calculating the fitness function based on the training results. This step represents the main action required for the subsequent assessment of the fitness function of each chromosome in the population. For each artificial neural network, there are training data, which depend on the problem being solved. The training data were obtained on the basis of a real technological process involving the induction brazing of waveguide paths of spacecraft.The assessment of the fitness of the chromosomes of the current population involves an assessment of the parameters of artificial neural networks associated with the problem of determining the best structure. Most often, such parameters are the recognition error.Checking the condition for stopping the algorithm. The work of the genetic algorithm stops after the expiration of the epochs allocated for training.The selection of chromosomes. In this study, tournament selection is used.The application of genetic operators. In this study, a two-point cross is used, and the probability of mutation is 10%.Creating a new population.Choosing the “best” structure for the artificial neural network.

The problem of determining the best structure for the artificial neural network is a multi-criteria optimization problem. To solve this problem, it is expedient to use a genetic algorithm that has proven itself to be effective in solving such problems [36,37,38]. The Fonseca and Fleming’s MultiObjective Genetic Algorithm (FFGA) algorithm has been successfully used to find the best structure of an artificial neural network. FFGA has one of the best convergences among other multiobjective evolutionary algorithms (MOEAs) [39], requires less computational costs [40], and solutions rarely go beyond the Pareto domain [41]. An additional advantage of FFGA is its natural character, in which the concept of Pareto dominance is applied directly in the very essence of the algorithm [35]. The algorithm is optimized by such parameters as the recognition error of the neural network model as well as the computational complexity of the model in the process of its direct operation [42,43,44].

Equation (1), providing a formal statement of the problem of determining the best ANN structure, can be written as follows:(1)$$\{\;\begin{array}{c} Err\left(M_{c},\;M_{w},\bar{act}\right)\to min\;\\ Cmp\left(M_{c},\bar{\;act}\;\right)\;\to min \end{array}$$
where *Err* is the total root mean square error (RMS) of the artificial neural network training, *Cmp* is the computational complexity of the neural network model, expressed in floating-point operations per second, *M_c_* is the artificial neural network connection matrix, *M_w_* is the matrix of weights of connections of an artificial neural network, and $\bar{\;act}$ is the vector of activation functions of neurons in an artificial neural network.

Depending on the type of problem, which is solved using the FFGA, the first RMS criterion in Equation (1) differs.

For the issue of identifying measurement errors, the criterion can be written as Equation (2):
(2)$$Err\;=\;\sqrt{\frac{1}{n}\cdot \sum_{i=1}^{n}\frac{{\left({\mathrm{Pr}}_{i}\;-\;{\mathrm{Pr}}_{i}^{\mathrm{real}}\right)}^{2}\;+\;{\left({\mathrm{Corr}}_{i}\;-\;{\mathrm{Corr}}_{i}^{\mathrm{real}}\right)}^{2}}{2}}$$
where Pr*_i_*, Corr*_i_* are values of ANN_ident_ output classes (Figure 6), and ${\mathrm{Pr}}_{i}^{\mathrm{real}},\;{\mathrm{Corr}}_{i}^{\mathrm{real}}$ are the real values of the appropriate classes.

For the issue of correction of measurement errors, the criterion can be written as Equation (3):
(3)$$Err\;=\;\sqrt{\frac{1}{n}\cdot \sum_{i=1}^{n}T_{i}^{\mathrm{corr}}\;-\;T_{i}^{\mathrm{real}}}$$
where $T_{i}^{\mathrm{corr}}$ are output values of ANN_corr_ (Figure 7), and $T_{i}^{\mathrm{real}}$ are real values of brazing temperature.

For the issue of measurement prediction, the criterion can be written as Equation (4):(4)$$Err\;=\;\sqrt{\frac{1}{n}\cdot \sum_{i=1}^{n}T_{i}^{\mathrm{predict}}\;-\;T_{i}^{\mathrm{real}}}$$
where $T_{i}^{\mathrm{corr}}$ are output values of ANN_predict_ (Figure 8).

For the issue of intelligent control, the criterion can be written as Equation (5):
(5)$$Err\;=\;\sqrt{\frac{1}{n}\cdot \sum_{i=1}^{n}\frac{{\left(\frac{W_{i}^{\mathrm{predict}}\;-\;W_{i}^{\mathrm{real}}}{W_{\max}}\right)}^{2}\;+\;{\left(\frac{h_{i}^{\mathrm{predict}}\;-\;h_{i}^{\mathrm{real}}}{h_{\max}}\right)}^{2}}{2}}$$
where $W_{i}^{\mathrm{predict}}$, $h_{i}^{\mathrm{predict}}$ are output values of ANN_control_ (Figure 9), $W_{i}^{\mathrm{real}}$, $h_{i}^{\mathrm{real}}$ are real values of the appropriate control variables, W_max_ is the maximum of the $W_{i}^{\mathrm{real}}$ set, and h_max_ is the maximum of the $h_{i}^{\mathrm{real}}$ set.

#### 2.4.2. Experimental Study on Choosing the Best Structure of an Artificial Neural Network

The optimization was conducted using the following hardware:Processor: AMD Ryzen 5 4500U, 2.38 GHz, 4 GHz peak, with six physical cores.RAM: 8 GB.512 GB solid-state drive, used as a swap file when needed.

The determination of the best structure of an artificial neural network for the identification of errors was carried out based on experimental studies. The experimental setup for determining the optimal structure of the ANN_ident_ was as follows:Number of hidden layers of the artificial neural network: from 1 to 10.Number of neurons per layer: 1 to 10.Number of individuals in the population: 30.Selection type: tournament.Number of individuals in the tournament: 5.Crossing type: uniform.Mutation probability: low.Maximum number of generations: 200.

The parameters of the artificial neural network are encoded in the genotype of the chromosome as follows:n—number of hidden layers.k_i_—number of neurons in layer i.

Multicriteria optimization is performed using the FFGA algorithm for the following parameters:General root–mean–square error of recognition—this parameter is minimized.Computational complexity in FLOPS—this parameter is minimized.

The parameters of the genetic algorithm were obtained experimentally. Such parameters as the number of layers and the number of artificial neurons per layer are limited to 10, based on the fact that the use of more complex structures greatly increases the computational complexity of the problem without a significant increase in the quality of the obtained solution. For example, with an increase in the number of neurons in a layer to 15, the complexity of the neural network model increases so that its computation time on the specified equipment exceeds 30 ms (due to the peculiarities of the measurement and control process in the induction brazing automation system).

Experimental studies, where the number of individuals in the population was varied from 10 to 100, showed that 30 is the most appropriate the number of individuals in terms of computational complexity and quality of the solution. The other parameters of the genetic algorithm were selected on the basis of the same provisions.

As a result of the selection of artificial neural network structures using FFGA, Pareto-optimal sets of neural network structures were obtained on the last generation of the genetic algorithm. Pareto-optimal fronts (Figure 11) make it possible to estimate the variety of formed neural network structures as well as to make a choice that is most effective from the point of view of the decision-maker.

Additionally, convergence graphs are obtained (Figure 12, Figure 13, Figure 14 and Figure 15), which illustrated the ANN training process for the following issues:Identifying measurement errors (Figure 12).Correction of measurement errors (Figure 13).Measurement prediction (Figure 14).Intelligent control (Figure 15).

The results show that the proposed approach has good convergence. The number of iterations of the algorithm required to find effective configurations of artificial neural networks is, depending on the problem, from 20 to 70.

In the case of the conducted experiment, the choice is made by the minimum total root mean square error. It is chosen because the computation complexity of all models is less than the required stated maximum. In case the model’s computation time is longer than required, it should be extracted from the considered set.

For example, for the issue of identifying measurement errors, the neural network with the following structure is obtained: 10-10-7-6-4-7-9-2. The network has six hidden layers, 10 neurons on the input layer, and two neurons on the output layer. The performance of this neural network model is 623 FLOPS, while the operating time of the model is 13 ms. The structure of the artificial neural network with the optimal structure for solving the problem of identifying non-standard errors is shown in Figure 16.

The ANN_ident_ with the selected optimal structure allows the problem of identifying errors in the measurement instruments in the process of the induction brazing of waveguide paths of spacecraft to be solved. The developed model makes it possible to recognize the presence of an error as well as to show the normality of the detected error. In cases where a standard error is found or there is no error, the process is controlled based on classical algorithms. In the case of the detection of a non-normative error, it is necessary to use ANN_corr_.

The implementation of the proposed technology will improve the control quality of the technological process of the induction brazing of waveguide paths of spacecraft by implementing:the identification of normative and non-normative errors of measurement instruments;the correction of non-standard errors, with the continuation of control based on classical algorithms;predicting measurements in the case of a failure of one of the pyrometric temperature measurement sensors, with the continuation of control based on classical algorithms;intelligent control in the event of a failure of both pyrometric sensors, which allows both to correctly complete the control of the technological process of induction brazing and to ensure a more even heating of the workpieces at the initial stage of heating. There are no readings from the pyrometric temperature sensors at this stage.

The developed artificial neural networks, with an experimentally selected optimal structure, had the following features:They allow for the identification of standard and non-standard errors of measurement instruments in the process of induction brazing, with an accuracy of 95.1%, while the operating time of the model is 13 ms.They allow for the correction of non-standard errors of measurement instruments in the process of the induction brazing of SCWP, with an accuracy of 96.6%, and the operating time of the model is 12 ms.They allow for the prediction of measurements in the process of the induction brazing of SCWP, with an accuracy of 96.7%, while the operating time of the model is 12.5 ms.They allow for the intelligent control of the SCWP induction brazing process in the absence of information about the process, with an accuracy of 94.5%, and the operating time of the model is 24 ms.

## 3. Results

### 3.1. Practical Implementation of Control Technology for the Induction Brazing of SCWP on the Basis of Intelligent Methods for Information Processing

The software products are developed as graphical applications designed for Windows. These products are compatible with all versions of Windows from Windows 7 to Windows 10.

The high-level programming language, Python, was chosen as the main tool. As a development methodology, the object-oriented approach was chosen [45,46,47].

The developed software is designed to solve the problem of determining the best structure of an artificial neural network when solving problems:The identification of errors of measurement instruments in the process of induction brazing SCWP.The correction of non-standard errors of measurement instruments in the process of the induction brazing of SCWP.The prediction of measurements in the process of induction brazing SCWP.Controlling the induction brazing of the SCWP in the absence of operational information about the heating temperatures of the elements of the SCWP assembly.

The software product implements the FFGA multicriteria genetic algorithm used to determine the best structure of an artificial neural network.

### 3.2. Designing a Software Control System for Induction Brazing SCWP Based on Intelligent Information Processing Methods

The purpose of the developed software is the intelligent control of the technological process of the induction brazing of waveguide paths of spacecraft. Algorithms for the identification and correction of errors of measurement instruments, an algorithm for predicting measurements, as well as an algorithm for the intelligent control of the technological process of induction soldering are implemented in the software. The software module consists of eight components (Figure 17).

The software has the following capabilities:Setting up pyrometric temperature measurement sensors.Setting up an artificial neural network for identifying errors in measurement instruments; correcting errors in measurement instruments; and predicting measurements; intelligent control of induction brazing, including training the network both from scratch and additional training.Starting the induction brazing control process.

In the initial stage of the intelligent control of the SCWP induction brazing process, data are obtained from pyrometric temperature sensors. If both sensors provide data on the process temperature, then errors are identified. In the absence of errors or in the presence of standard errors, the data are transmitted further for TP control based on classical control algorithms.

In the presence of non-standard errors, they are corrected using further data transmission for control based on classical algorithms. In the absence of readings from one of the pyrometers, the data from the second sensor are determined based on the forecast obtained using ANN_predict_, while the control is also based on classical control algorithms.

In the settings tab of the artificial neural networks (Figure 18) used in the SCWP induction brazing control system, there is access to training and additional training of artificial neural networks used to solve problems associated with the identification and correction of errors in the measurement instruments, predicting measurements, and control of the SCWP induction brazing process. Initially, the software already contains pre-trained models, but in this tab, one can both train each artificial neural network from scratch and perform additional training.

### 3.3. Experimental Study and Discussion

To study the efficiency of an intelligent control system for the induction brazing of waveguide paths, a number of experiments were carried out using experimental equipment. A graph with induction brazing control, in the absence of measurement errors, is shown in Figure 19.

In a series of experiments, the average operation time of the ANN_ident_ was found to be no more than 13 ms. As can be seen from the figure, the proposed control technology, as well as the software that implements this technology, provides a sufficiently high quality of control, since no overregulation is observed on the graph of the induction brazing process.

A graph presenting the induction brazing control in the case of correcting measurements using ANN_corr_ is shown in Figure 20.

In a series of experiments, the average operation time of the ANN_ident_ was no more than 13 ms and, for the ANN_corr_, no more than 12 ms. As can be seen in Figure 20, without the use of the technology of the intelligent control of induction brazing, before entering the stabilization stage, there is a sharp surge in temperature from the correcting pyrometer. This is caused by a non-standard error due to the melting of the flux, which the ANN_corr_ corrects, that provides a minimum discrepancy in the heating of the elements of the waveguide path assembly.

A graph presenting the induction brazing control in the case of modeling temperature readings using ANN_predict_, when the correction pyrometer fails is shown in Figure 21.

For a series of experiments, the average operation time of the ANN_ident_ was no more than 13 ms and, for the ANN_predict_, no more than 12.5 ms. After the failure of the correcting pyrometer, further control of the induction brazing process was conducted based on the data obtained using the ANN_predict_.

A graph with the induction brazing control using a neural network model (ANN_control_), with the simulation of the readings of the control and correcting pyrometers using an ANN_predict_, is shown in Figure 22.

For a series of experiments, the average operation time of the ANN_ident_ was no more than 13 ms; for the ANN_predict_, no more than 12.5 ms; and for the ANN_control_, no more than 24 ms. Figure 22 shows a graph of the process during which the correction and control pyrometers fail. After the failure of the correcting pyrometer, further control of the induction brazing process is conducted by the ANN_control_ based on the data obtained using the ANN_predict_ for the corrective and control pyrometers.

The quality of control in every experiment is ensured by the following:The absence of overregulation in the stage of process stabilization.The absence of a significant difference in heating temperatures between the brazed elements of the waveguide path assembly.The short duration of the induction brazing process, which has a positive effect on the quality of the products, since the heating time is reduced.

The results of the experiments (Figure 19, Figure 20, Figure 21 and Figure 22) clearly confirm the achievement of the tasks set:The method of identification and correction of the errors of measurement instruments in the process of induction brazing, developed by the authors, allows the influence of non-standard errors of the measurements of pyrometric sensors to be reduced.The method proposed by the authors for predicting measurements of pyrometric sensors makes it possible to perform the induction brazing of SCWP, achieving the required quality under conditions of incomplete information.The intelligent control algorithm for the induction brazing of the SCWP developed by the authors allows for the control of the induction brazing of the SCWP, achieving the required quality in the absence of information from pyrometric sensors.The developed software makes it possible to improve the control quality of the technological process of induction brazing under conditions of incomplete or unreliable information from pyrometric sensors.

Based on the results of all experiments, high-quality brazed joints were obtained. As can be seen in Figure 22, at 19 s, the control switches from the classic PID controller to the ANN_control_. Based on the results of the experiments, it can be concluded that the intelligent control presented in this study allows the discrepancy between the brazing process and the heating program to be compensated for in a short time and the discrepancy in the temperatures of the elements to be brazed, and it ensures a uniform flow of solder along the entire perimeter of the joint.

Based on the results of all the experiments, high-quality brazed joints were obtained. Table 1 shows a comparison of the control quality of the induction brazing of the SCWP without the use of intelligent methods of information processing and using intelligent methods of information processing.

Microsections of the brazed waveguides obtained from the experimental results are shown in Figure 23, from which, it can be seen that, according to the results of the experiments, high-quality brazed joints were obtained. It can be seen that there was a high-quality flow of solder throughout the entire soldering area, which ensures high-quality products.

Based on the results of the experiments, it can be concluded that the formed intelligent control allows the discrepancy between the brazing process and the heating program to be compensated for in a short time and the discrepancy in the temperatures of the elements to be brazed, and it ensures a uniform flow of solder along the entire perimeter of the joint.

## 4. Conclusions

In this study, an intelligent technology is developed for controlling the induction brazing of waveguide paths of spacecraft. The paper presents algorithmic and model support for this technology. The efficiency of the proposed solution has been tested through experiments using a prototype of the experimental equipment. In all the experiments, high-quality brazed joints were obtained. The use of the intellectual technology presented in this study will improve the control quality of the technological process of the induction brazing of waveguide paths of spacecraft and, therefore, improve the quality of the products manufactured in enterprises of the rocket and space industry.

Based on the verification results, it can be concluded that the developed software allows for the control of the brazing process, both under normal conditions and under conditions of incomplete information caused by a failure of one or two pyrometers.

The experimental research has shown that the quality of management is ensured by:The absence of overregulation in the stage of process stabilization.The absence of a significant difference in the heating temperatures between the brazed elements of the assembly of the waveguide path.The short duration of the induction brazing process, which has a positive effect on the quality of the products, since the heating time is reduced.

As directions for further research, the following should be noted:Designing a model of artificial neural networks for an indefinite number of measurement instruments.Developing intelligent technology for use in the case of different types of measurement instruments.Adapting and applying a team of intelligent technologies using other types of artificial neural networks, for example, convolutional neural networks for processing visual information about the technological process of induction brazing and Kohonen maps for clustering a variety of different measuring devices.Applying explainable artificial intelligence methods to implement reverse inference in order to explain the causes of an automated system for a process unit operator.

## Figures and Tables

**Figure 1 sensors-21-04199-f001:**
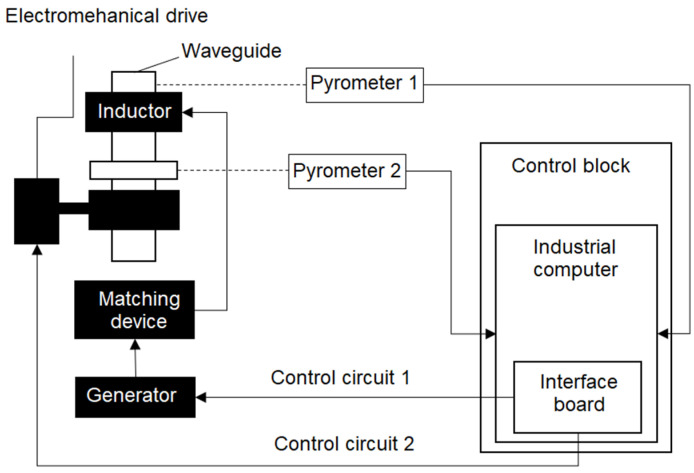
Functional diagram of the induction brazing unit.

**Figure 2 sensors-21-04199-f002:**
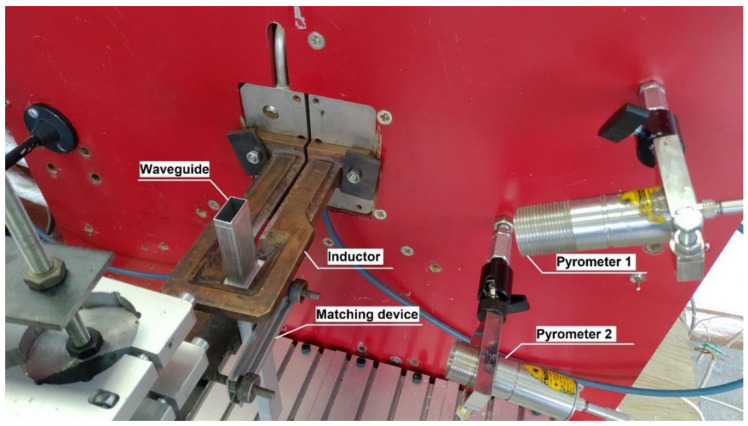
Induction brazing installation.

**Figure 3 sensors-21-04199-f003:**
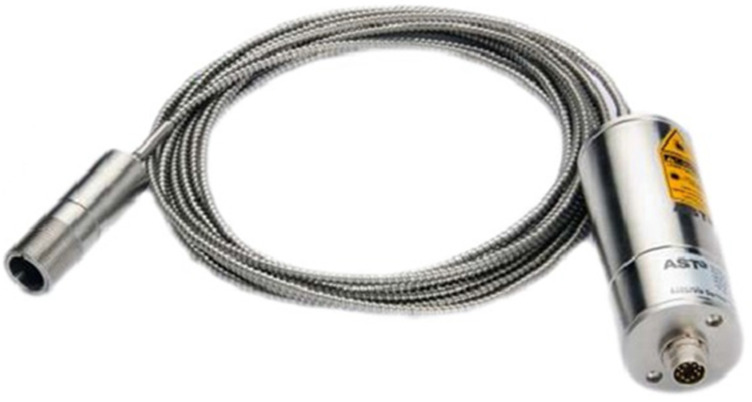
Pyrometer AST 250.

**Figure 4 sensors-21-04199-f004:**
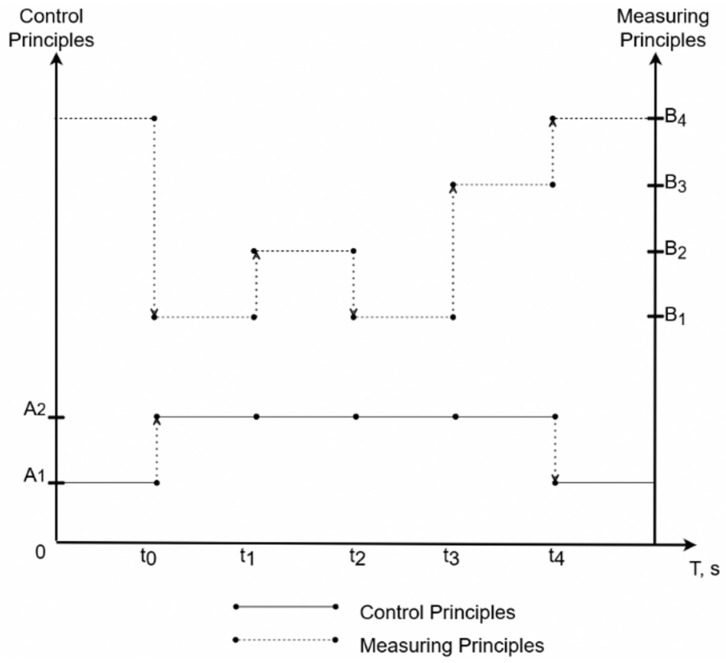
Scenario diagram of the proposed technology, where A1—intelligent control, A2—control according to classical algorithms, B1—direct measurements from pyrometers, B2—measurements corrected using ANN, B3—measurements predicted using ANN, and B4—measurements from pyrometers are absent.

**Figure 5 sensors-21-04199-f005:**
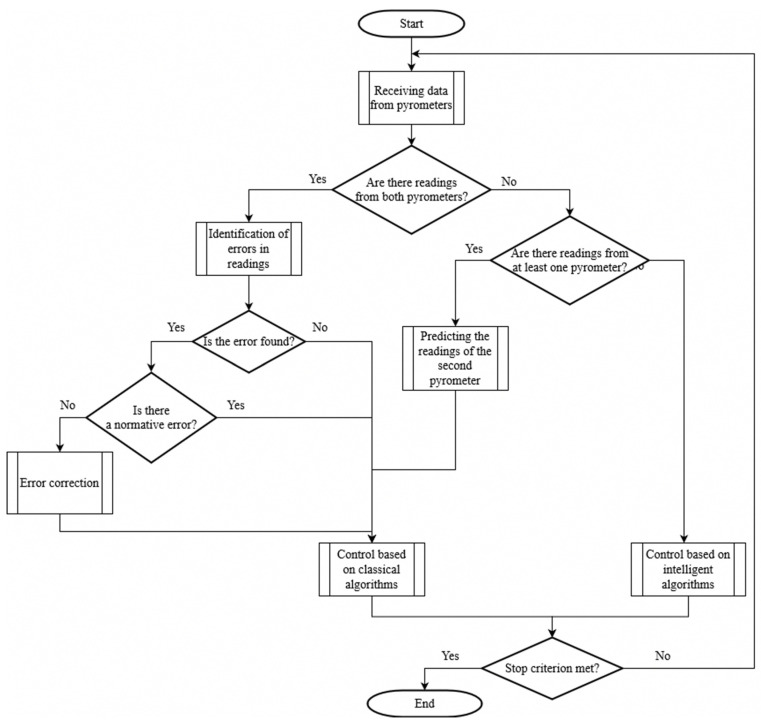
Algorithm for the intelligent control of the induction brazing process.

**Figure 6 sensors-21-04199-f006:**
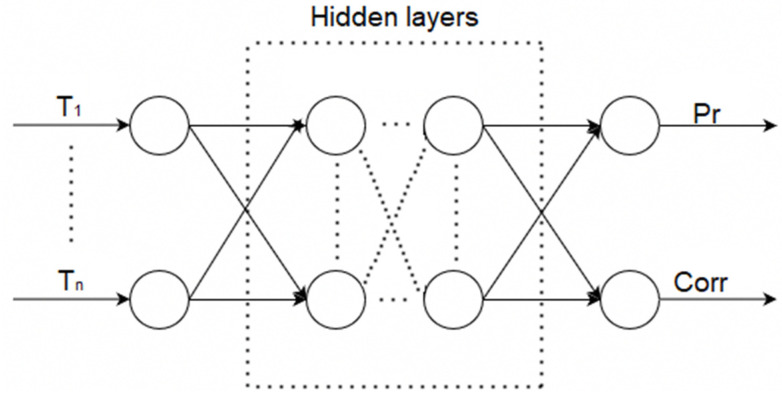
General structure of an artificial neural network for the identification of errors, where T_1_, ..., T_n_ are input data, which are time-series data from the brazing process, i.e., the temperature of one of the brazed elements, Pr is the output class, indicating the presence of an error, and Corr is the output class, indicating the normality of the error.

**Figure 7 sensors-21-04199-f007:**
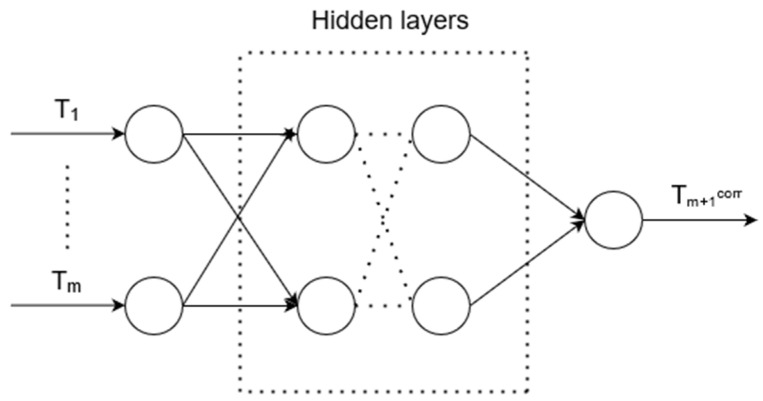
Artificial neural network structure for error correction, where T_1_, ..., T_m_ are input data representing time-series data from the brazing process, i.e., the temperature of one of the brazed elements, and T_m+1_^corr^ is the output value, which is a corrected measurement.

**Figure 8 sensors-21-04199-f008:**
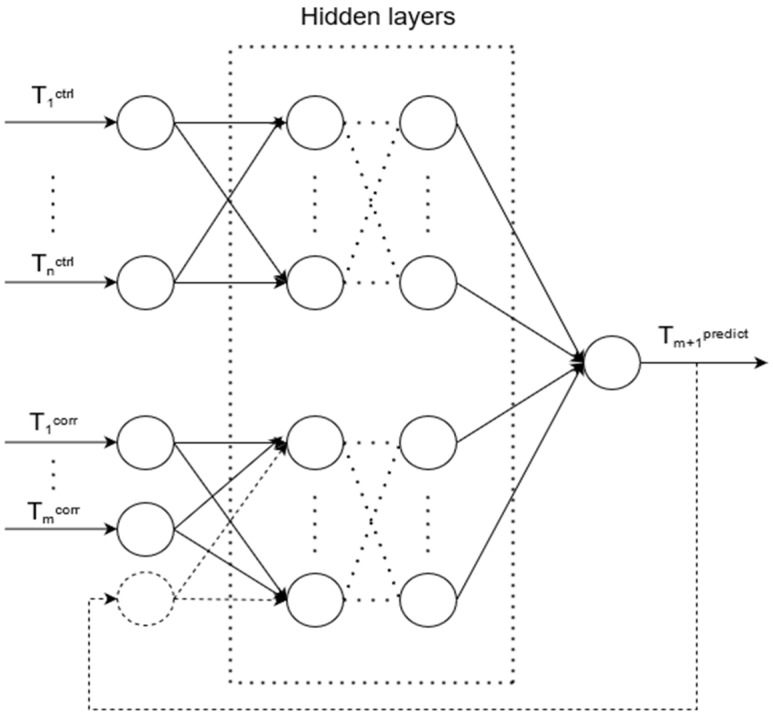
Artificial neural network structure for predicting measurements, where T_1_^ctrl^, ..., T_m_^ctrl^ are input data representing time-series data from the control pyrometer, T_1_^corr^, ..., T_m_^corr^ are input data representing time-series data from the control pyrometer, and T_m+1_^predict^ is an output value representing the predicted measurement. Over time, the forecasting accuracy will gradually decrease, since the proportion of predicted rather than actual values in the input data will increase.

**Figure 9 sensors-21-04199-f009:**
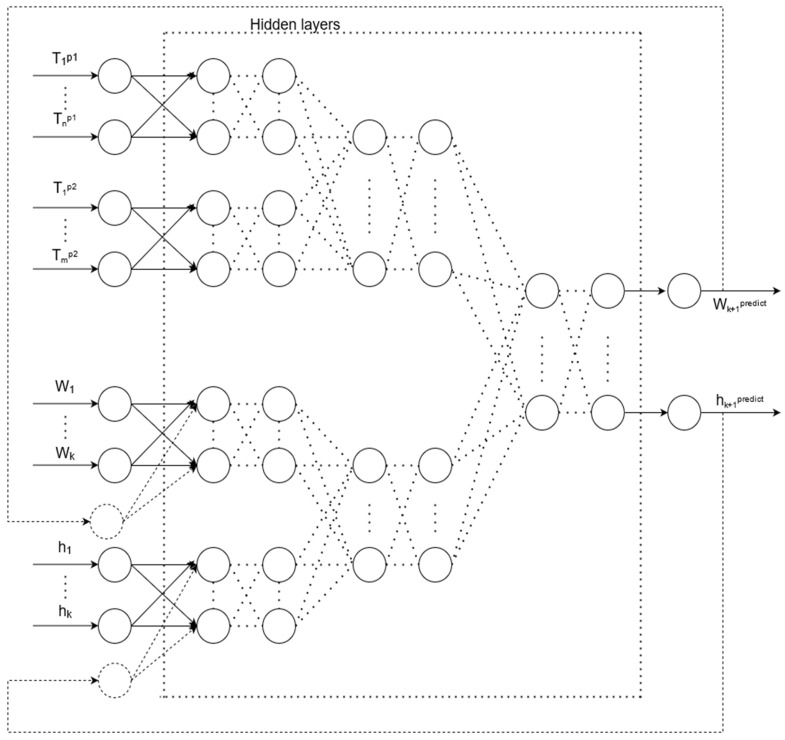
Artificial neural network structure for predicting measurements, where T_1_^p1^, ..., T_n_^p1^ are input data representing time-series measurements of the first pyrometer, T_1_^p2^, ..., T_m_^p2^ are input data representing time-series measurements of the second pyrometer, h_1_, ..., h_k_ are input data representing the time series of values of the distance from the inductor to the assembly of the waveguide path, W_1_, ..., W_k_ are input data, which are the time series of power settings, h_k+1_ are the output of ANN, which is the calculated value of the distance from the inductor to the workpiece, and W_k+1_ are the output of ANN, which is the calculated value of the inductor power setting.

**Figure 10 sensors-21-04199-f010:**
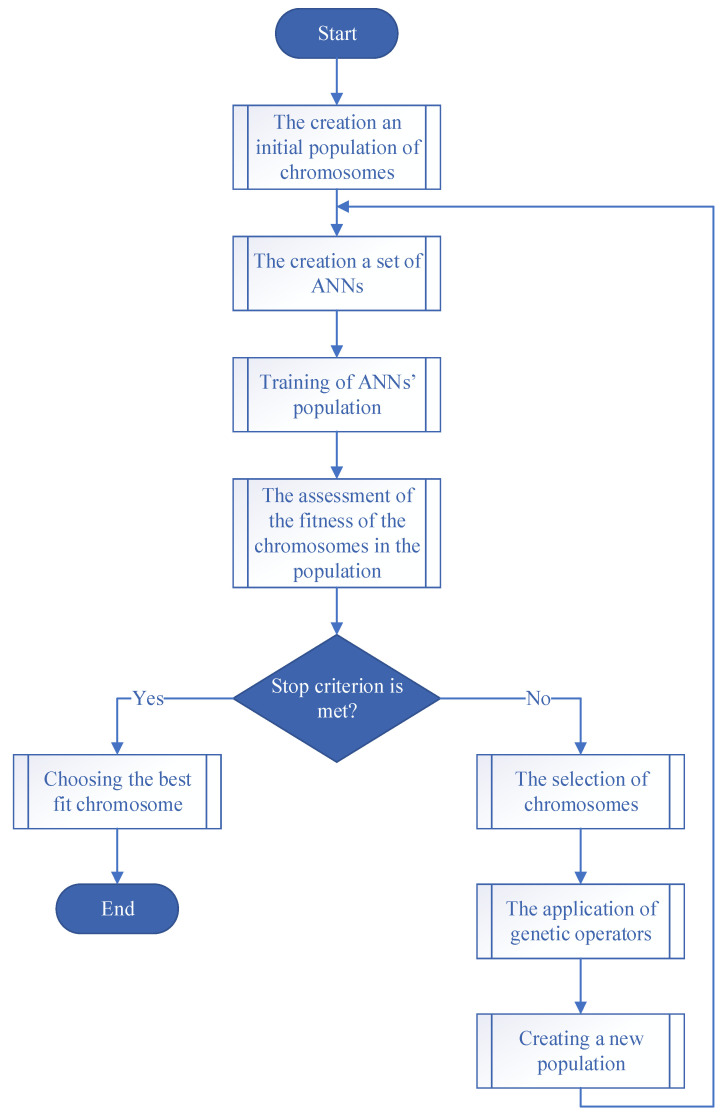
Block diagram of a genetic algorithm for determining the optimal structure of an artificial neural network.

**Figure 11 sensors-21-04199-f011:**
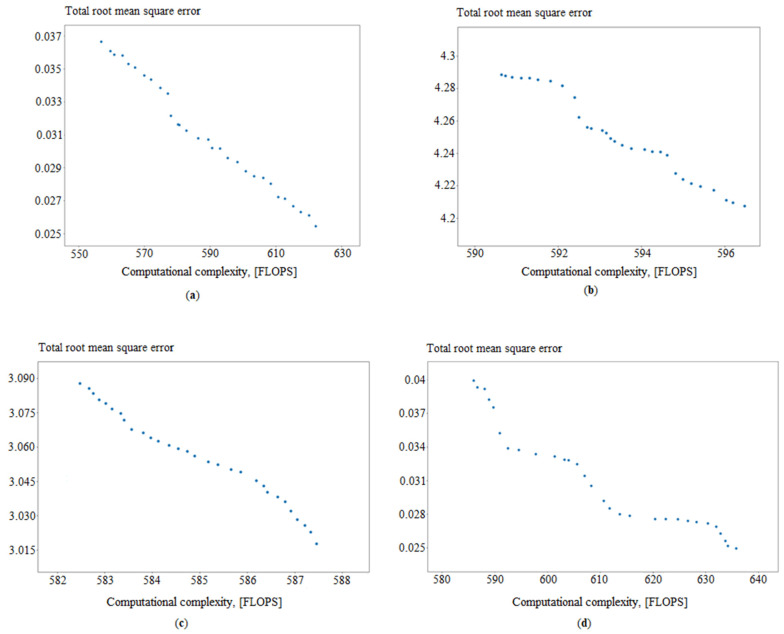
Approximations of the Pareto-optimal front on the last generation in determining the best from, where (**a**) ANN_ident_, (**b**) ANN_corr_, (**c**) ANN_predict_, and (**d**) ANN_control_.

**Figure 12 sensors-21-04199-f012:**
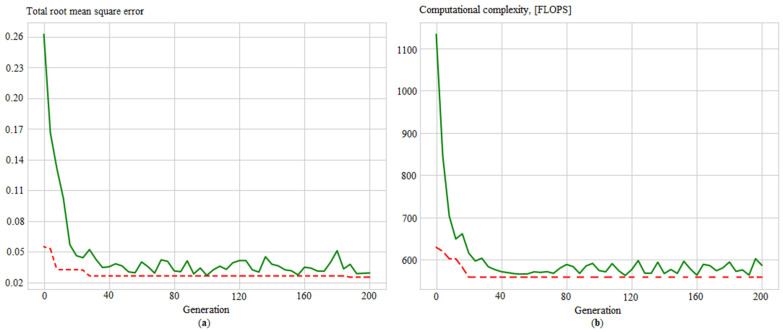
Evolution of the ANN_ident_ in the process of determining its best structure by: (**a**) total root mean square error and (**b**) computational complexity, where solid green lines are average values per generation, and dotted red lines are minimum values per generation.

**Figure 13 sensors-21-04199-f013:**
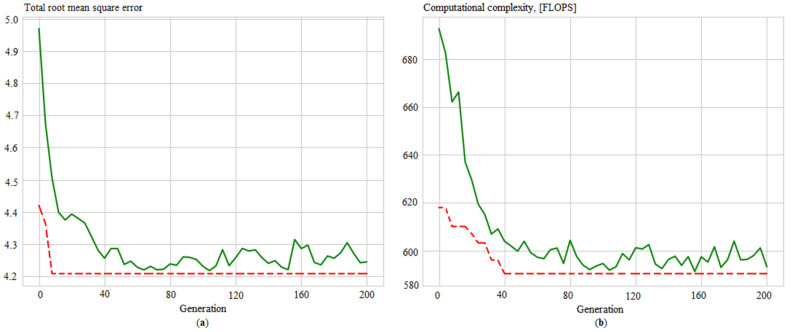
Evolution of the ANN_corr_ in the process of determining its best structure by (**a**) total root mean square error; (**b**) computational complexity, where solid green lines are average values per generation, and dotted red lines are minimum values per generation.

**Figure 14 sensors-21-04199-f014:**
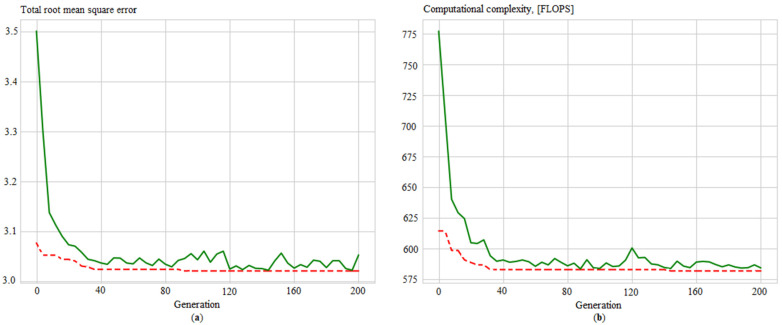
Evolution of the ANN_predict_ in the process of determining its best structure by: (**a**) total root mean square error; (**b**) computational complexity, where solid green lines are average values per generation, and dotted red lines are minimum values per generation.

**Figure 15 sensors-21-04199-f015:**
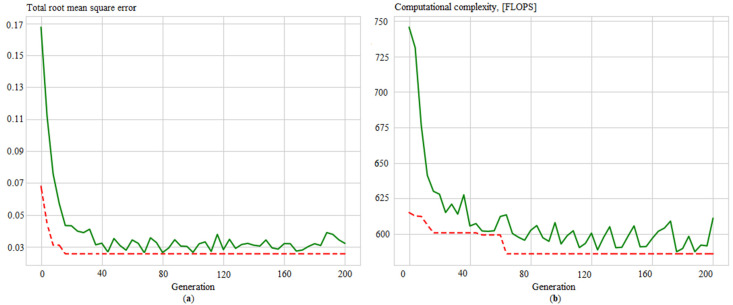
Evolution of the ANN_control_ in the process of determining its best structure by: (**a**) total root mean square error; (**b**) computational complexity, where solid green lines are average values per generation, and dotted red lines are minimum values per generation.

**Figure 16 sensors-21-04199-f016:**
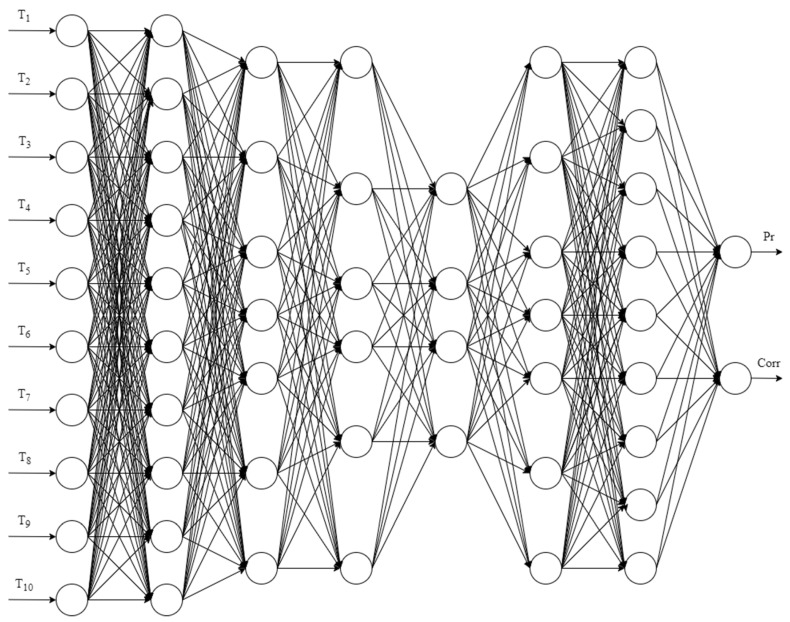
The structure of the artificial neural network for the identification of errors with the optimal structure.

**Figure 17 sensors-21-04199-f017:**
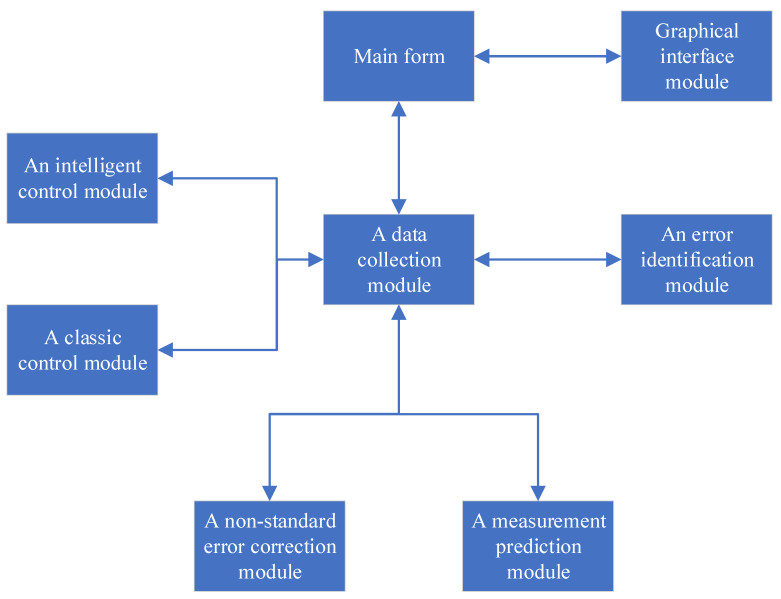
Structural diagram of the software product.

**Figure 18 sensors-21-04199-f018:**
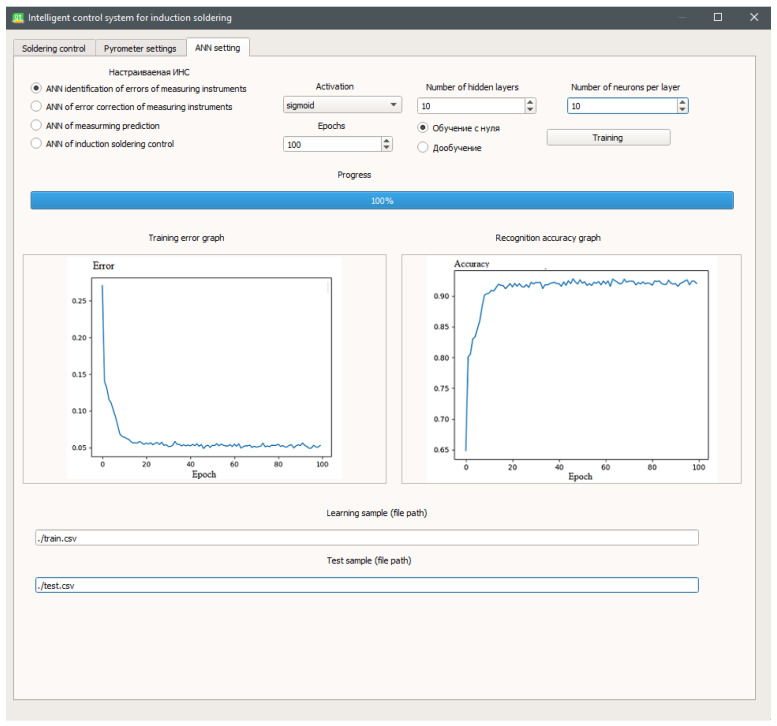
Artificial neural networks settings tab.

**Figure 19 sensors-21-04199-f019:**
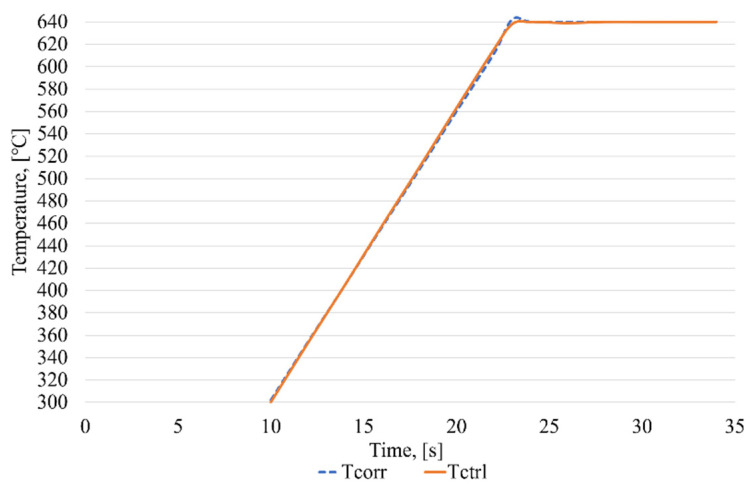
Induction brazing control process graph, where Tcorr—temperature values obtained from a correcting pyrometer, and Tctrl—temperature values obtained from a control pyrometer.

**Figure 20 sensors-21-04199-f020:**
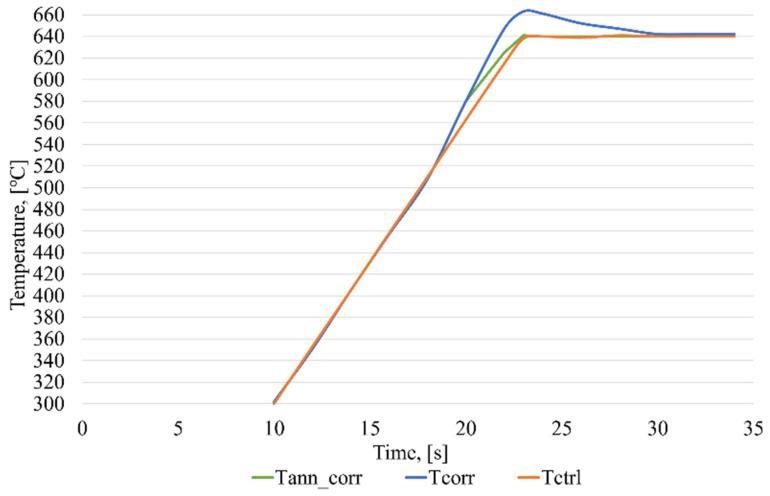
Graph of the intelligent control of the induction brazing process in the case of correcting the measurements from one of the pyrometers, where Tcorr—temperature values obtained from a correcting pyrometer, Tctrl—temperature values obtained from a control pyrometer, and Tann_corr—temperature values corrected using ANN_corr_.

**Figure 21 sensors-21-04199-f021:**
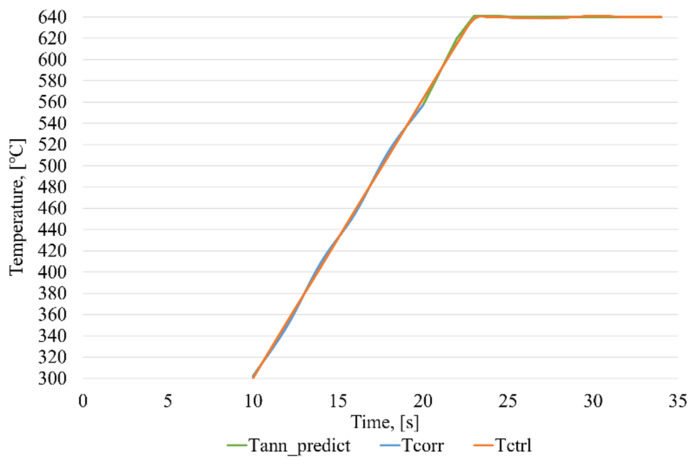
Graph of the intelligent control of the induction brazing process in the case of simulating the temperature measurements from one of the pyrometers, where Tcorr—temperature values obtained from a correcting pyrometer, Tctrl—temperature values obtained from a control pyrometer, Tann_predict—temperature readings from a pyrometer, modeled using ANN_predict_.

**Figure 22 sensors-21-04199-f022:**
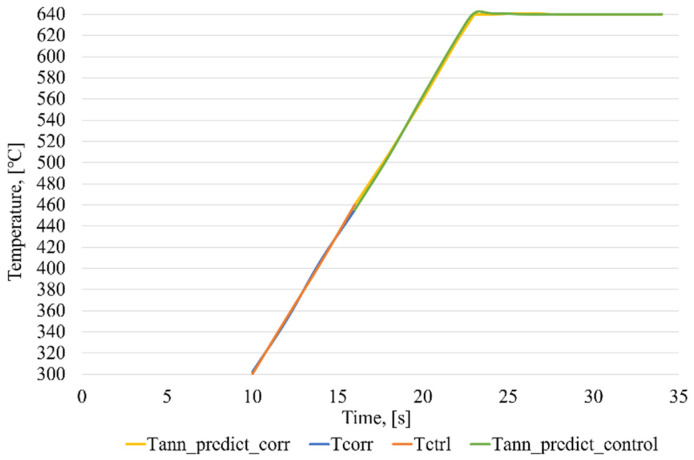
Graph of the intelligent control of the induction brazing process in the case of simulating the measurements of the control and correcting pyrometers, with the control based on the artificial neural network, where Tcorr—temperature values obtained from a correcting pyrometer, Tctrl—temperature values obtained from a control pyrometer, Tann_corr—temperature values corrected using ANN_corr_, Tann_predict—temperature readings from a pyrometer, modeled using ANN_predict_, Tann_predict_corr—simulated temperature readings from a correction pyrometer, Tann_predict_control—simulated temperature readings from the control pyrometer.

**Figure 23 sensors-21-04199-f023:**
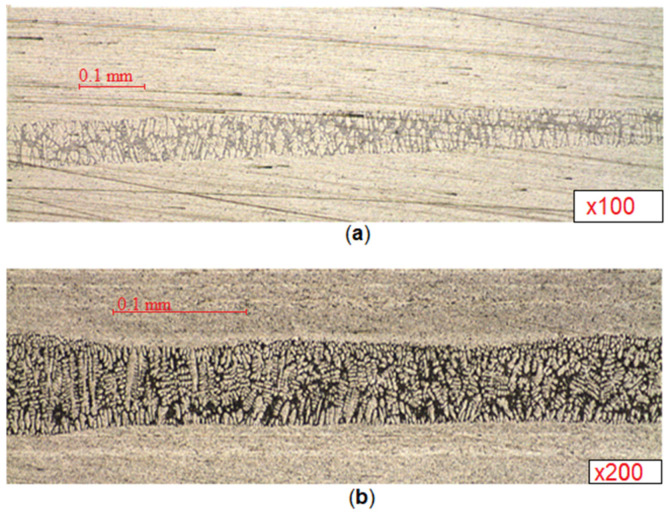
Microsections of the brazed waveguides: (**a**) microsection of a pipe-flange waveguide assembly; (**b**) microsection of a pipe-coupling waveguide assembly.

**Table 1 sensors-21-04199-t001:** Quality comparison of the induction brazing control principles.

Comparison Criteria	Classic Technology	Intelligent Technology
Overregulation (%)	0–20	0
Difference in heating temperatures of the elements of the SCWP assembly to be brazed (°C)	20–100	0–10
Duration of the SCWP induction brazing process (sec)	20–60	30–35

## Data Availability

Not applicable.

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
