# Peer review of "The Use of Collections of Artificial Neural Networks to Improve the Control Quality of the Induction Soldering Process"

_sensors, 2021, doi:10.3390/s21124199_

Round 1

Reviewer 1 Report

- it is important to mention that the authors due mention the transfer function applied in each architecture presented.
- also, mention if the input and output variables are normalized?
- Figure 1 is unnecessary.
- In lines 141-152, where the authors mention: "Mathematical models based on artificial neural networks have been successfully used to solve various problems. Examples of successful use of artificial neural networks are:" add in other line:  The use of ANN to solve energy problems (Reference: The multivariable inverse artificial neural network combined with GA and PSO to improve the performance of solar parabolic trough collector. Applied Thermal Engineering Volume 189, 5 May 2021, 116651;
and Control scheme formulation for a parabolic trough collector using inverse artificial neural networks and particle swarm optimization. J Braz. Soc. Mech. Sci. Eng. 43, 176 (2021)). 

Author Response

Dear Reviewer 1,

Thank you for your positive comments on our paper. You provided remarks about it, so we’ve made changes in the paper.

Remark 1: it is important to mention that the authors due mention the transfer function applied in each architecture presented.

Reply: Thank you for the remark. A leaky linear rectifier (Leaky ReLU) was chosen as the activation (also transfer) function for each architecture presented. We have added paragraph about it (lines 268-273).

Remark 2: also, mention if the input and output variables are normalized?

Reply: Thank you for the comment. Input and output variables are normalized. We have added paragraph about it (lines 268-273).

Remark 3: Figure 1 is unnecessary.

Reply: Thank you for the comment. We have made correction (lines 96-139).

Remark 4: In lines 141-152, where the authors mention: "Mathematical models based on artificial neural networks have been successfully used to solve various problems. Examples of successful use of artificial neural networks are:" add in other line:  The use of ANN to solve energy problems (Reference: The multivariable inverse artificial neural network combined with GA and PSO to improve the performance of solar parabolic trough collector. Applied Thermal Engineering Volume 189, 5 May 2021, 116651;

and Control scheme formulation for a parabolic trough collector using inverse artificial neural networks and particle swarm optimization. J Braz. Soc. Mech. Sci. Eng. 43, 176 (2021)).

Reply: Thank you for the comment. We have now added other line with references (lines 125).

With best regards,

Dr. Vadim Tynchenko

Reviewer 2 Report

The paper is well-written and original. In general, I think the paper is worth publishing. However, I have a couple of comments: 

1) the paper contains a lot of information that is not related or can be expected to be known to the audience. Specifically: lines 53-75 contain general NN related info that is not necessary to understand this paper. Section 2.1 is quite basic and introduces NN. You can expect that to be known. Section 2.5 is standard GA introduction the reader will likely know. This section can be abbreviated or removed without any loss in content. The selection of Python as an implementation language is not contributing to the depth of the paper (section 3.1). 

2) Figure 4 is hard to read as the contrast between text and background is not strong enough. It is very hard to read in black and white print. 

3) Figure 6 is good but could use some annotation. There is a description in the text but one really needs to read the text to understand the figure. 

4) The rationale of choice for using NN is not sufficient. You may have chosen Fuzzy logic or other optimised control technologies. Literature will help. The expectations and requirements of a solutions should be mentioned first and then the technical solution determined on that basis. NN have also interpretation and explanation issues which other model based techniques do not have. I suggest making a stronger point for the choice of NN.

5) The FFGA needs a bit more rationale of choice. There are so many other MOEA out there. 

6) I would have expected some more information about the FPGA section, such as convergence graphs, NN structure evolution, runtime and rationale for parameter settings (e.g. a population of 30 appears rather low), variations and average quality of the solutions, robustness of the solution (do small changes in the solution have large effects or cause instabilities).

7) The figures 15 and 16 are implementation screenshots and the whole sections 3.1 and 3.2 are quite implementation related rather than contributing to academic knowledge. This could be abbreviated. 

8) The text in sections 3.3 is very repetitive almost to the level of individual words. This is exhausting the reader. I suggest focussing on the main contributions to knowledge. 

9) Table 1 is clearly reaching into the margins. 

10) Figure 22 is missing a scale and annotations.
      You may use the figures and material analysis to support the aim of this research. 

Author Response

Dear Reviewer 2,

Thank you for your positive comments on our paper. You provided remarks about it, so we’ve made changes in the paper.

Remark 1: the paper contains a lot of information that is not related or can be expected to be known to the audience. Specifically: lines 53-75 contain general NN related info that is not necessary to understand this paper. Section 2.1 is quite basic and introduces NN. You can expect that to be known. Section 2.5 is standard GA introduction the reader will likely know. This section can be abbreviated or removed without any loss in content. The selection of Python as an implementation language is not contributing to the depth of the paper (section 3.1).

Reply: Thank you for the remark. We have changed and removed the information which can be expected to be known to the audience. Lines 52-55, 94-97, 102-112, 113-125, 355-363, 428-443,569-570.

Remark 2: Figure 4 is hard to read as the contrast between text and background is not strong enough. It is very hard to read in black and white print.

Reply: Thank you for the comment. We have now remade the figure. Lines 145.

Remark 3: Figure 6 is good but could use some annotation. There is a description in the text but one really needs to read the text to understand the figure.

Reply: Thank you for the comment. We have now added annotation (lines 203-206).

Remark 4: The rationale of choice for using NN is not sufficient. You may have chosen Fuzzy logic or other optimised control technologies. Literature will help. The expectations and requirements of a solutions should be mentioned first and then the technical solution determined on that basis. NN have also interpretation and explanation issues which other model based techniques do not have. I suggest making a stronger point for the choice of NN.

Reply: Thank you for the comment. We have now added a stronger point for the choice of NN (lines 102-108).

Remark 5: The FFGA needs a bit more rationale of choice. There are so many other MOEA out there.

Reply: Thank you for the comment. We have added paragraph about it (lines 409-420).

Remark 6: I would have expected some more information about the FPGA section, such as convergence graphs, NN structure evolution, runtime and rationale for parameter settings (e.g. a population of 30 appears rather low), variations and average quality of the solutions, robustness of the solution (do small changes in the solution have large effects or cause instabilities).

Reply: Thank you for the comment. We have added more information about FFGA section. Lines 468-523.

Remark 7: The figures 15 and 16 are implementation screenshots and the whole sections 3.1 and 3.2 are quite implementation related rather than contributing to academic knowledge. This could be abbreviated.

Reply: Thank you for the comment. We removed figures 15 and 16(lines 614-616) and abbreviated sections 3.1 and 3.2 (lines 566-581, 584-616).

Remark 8: The text in sections 3.3 is very repetitive almost to the level of individual words. This is exhausting the reader. I suggest focussing on the main contributions to knowledge.

Reply: Thank you for the comment. We have changed and removed repetitive text and focusing on the main contributions to knowledge. Lines 625-629, 637-643, 651-654, 666-677

Remark 9: Table 1 is clearly reaching into the margins.

Reply: Thank you for the comment. We have corrected table 1 (line 704).

Remark 10: Figure 22 is missing a scale and annotations.

Reply: Thank you for the comment. We have added a scale and annotations (lines 715-717).

With best regards,

Dr. Vadim Tynchenko

Round 2

Reviewer 1 Report

Thanks for the changes